# A chiral selectivity relaxed paralog of DTD for proofreading tRNA mischarging in Animalia

Santosh Kumar Kuncha[1,2], Mohd Mazeed[1], Raghvendra Singh[1], Bhavita Kattula[1], Satya Brata Routh[1] & Rajan Sankaranarayanan [1]

D-aminoacyl-tRNA deacylase (DTD), a bacterial/eukaryotic *trans*-editing factor, removes D-amino acids mischarged on tRNAs and achiral glycine mischarged on tRNA[Ala]. An invariant cross-subunit Gly-*cis*Pro motif forms the mechanistic basis of L-amino acid rejection from the catalytic site. Here, we present the identification of a DTD variant, named ATD (Animalia-specific tRNA deacylase), that harbors a Gly-*trans*Pro motif. The *cis*-to-*trans* switch causes a "gain of function" through L-chiral selectivity in ATD resulting in the clearing of L-alanine mischarged on tRNA[Thr](G4•U69) by eukaryotic AlaRS. The proofreading activity of ATD is conserved across diverse classes of phylum Chordata. Animalia genomes enriched in tRNA[Thr](G4•U69) genes are in strict association with the presence of ATD, underlining the mandatory requirement of a dedicated factor to proofread tRNA misaminoacylation. The study highlights the emergence of ATD during genome expansion as a key event associated with the evolution of Animalia.

[1] CSIR–Centre for Cellular and Molecular Biology, Uppal Road, Hyderabad 500007, India. [2] Academy of Scientific and Innovative Research (AcSIR), CSIR–CCMB Campus, Uppal Road, Hyderabad 500007, India. Santosh Kumar Kuncha and Mohd Mazeed contributed equally to this work. Correspondence and requests for materials should be addressed to R.S. (email: sankar@ccmb.res.in)

Translational quality control is a complex and tightly regulated process which involves editing of errors in most scenarios. However, it also encompasses a targeted and selective compromise in fidelity, thereby allowing percolation of errors under specific conditions such as oxidative stress. It ensures an optimum dynamic balance in the cellular proteome and hence overall cellular homeostasis. A multitude of factors—from aminoacyl-tRNA synthetases (aaRSs) to ribosome, as well as proteasome—play significant roles in performing this complex phenomenon[1–7]. A key step in this process includes decoupling of D-amino acids mischarged on tRNAs. This function, termed "chiral proofreading", is performed by a dedicated *trans*-editing factor called D-aminoacyl-tRNA deacylase (DTD)[8–11]. The chiral proofreading enzyme forms one of the major cellular checkpoints, which also includes aaRSs, elongation factor-Tu (EF-Tu) and ribosome, to prevent infiltration of D-amino acids into translational machinery[12–19].

DTD—present throughout Bacteria (except cyanobacteria) and Eukarya—is a homodimeric enzyme which has two active sites located at the dimeric interface. A crucial component of each of the active sites is an invariant Gly-*cis*Pro dipeptide motif belonging to one monomer, which is inserted into active site of the other monomer. DTD employs this cross-subunit Gly-*cis*Pro motif to ensure substrate stereospecificity[11]. The architecture of DTD's chiral proofreading site is such that it sterically excludes even the smallest amino acid with L-chirality, viz., L-alanine. Thus, strict L-chiral rejection rather than D-chiral selection forms the only mechanistic basis of DTD's enantioselectivity[20]. Interestingly, in archaea and cyanobacteria, chiral proofreading is performed by DTD2 and DTD3, respectively; the latter two are non-homologous to DTD[21,22]. However, in archaea, which lack DTD, a DTD-like module is covalently appended to threonyl-tRNA synthetase (ThrRS) as the N-terminal domain (NTD) that edits L-serine misacylated on tRNA^Thr[23–26]. Thus, both DTD-like fold (comprising DTD and NTD) and chiral proofreading function (performed by DTD, DTD2, and DTD3) are conserved across all domains of life. Biochemically, the DTD-like fold is an RNA-based catalyst that employs only the 2′-OH of adenosine-76 (A76) at the 3′-terminus of tRNA rather than protein side chains for catalysis at the RNA–protein interface[20,27].

Proofreading during aminoacyl-tRNA synthesis has been proposed and extensively studied so far in the context of errors only in amino acid selection by aaRSs[28–38]. Defects in proofreading have been associated with multiple cellular pathologies including neurodegeneration in mouse and cell death[23,39–50]. Amino acids are substantially smaller in size compared to tRNAs, and are also similar in structure/chemistry in several cases. Consequently, errors in amino acid selection by synthetases are

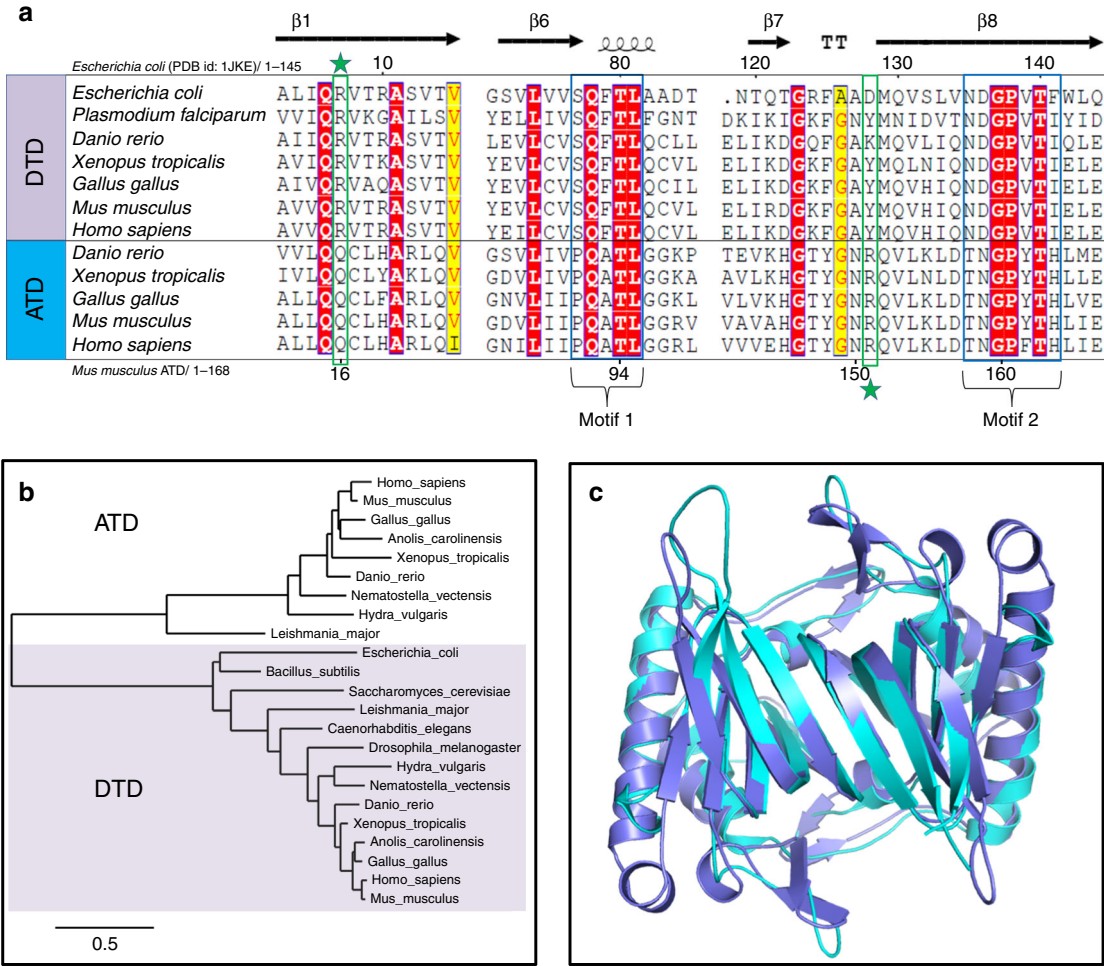

**Fig. 1** ATD is a variant of DTD. **a** Multiple sequence alignment showing similar but distinct and characteristic sequence motifs in DTD and ATD (motifs 1 and 2). The highly conserved arginine in DTD (Arg7, EcDTD) is indicated by a star above, whereas the invariant arginine in ATD (Arg151, MmATD) is highlighted by a star below. **b** Phylogenetic classification of DTD and ATD showing their grouping into two separate categories. **c** Crystal structure of MmATD homodimer (blue) superimposed on that of PfDTD homodimer (cyan; PDB id: 4NBI)

significantly higher (about one in $10^3$–$10^2$) than the overall error observed during translation of the genetic code (about one in $10^4$–$10^3$), thereby necessitating evolution of various *cis*- and *trans*-editing activities associated with nearly half of the known aaRSs[37,38,51]. Errors in tRNA selection, which either happen naturally and constitutively or are induced by environmental conditions (such as oxidative/temperature/antibiotic stress), have been noted in several instances, although such mistakes are not as common as those in amino acid selection[7]. However, dedicated proofreading factors for correcting mistakes in tRNA selection have not been reported till date.

Here, we describe the identification and characterization of a unique DTD-like factor, named Animalia-specific tRNA deacylase (ATD). Present in kingdom Animalia and more specifically all across phylum Chordata, ATD proofreads a critical tRNA selection error made by AlaRS. An unprecedented switch from the chiral-selective "Gly-*cis*Pro" dipeptide in DTD to "Gly-*trans*-Pro" in ATD is the key to ATD's gaining of L-chiral selectivity. This "gain of function" through relaxation of substrate chiral specificity underlies ATD's unique capability of correcting the error in tRNA selection. The strict coexistence of the proofreading factor with the error-inducing tRNA species underlines its requirement for translational quality control in Animalia. Our study represents the identification of a proofreading factor that is responsible for the correction of an error in tRNA selection during translation of the genetic code.

## Results

**Subtle differences in characteristic motifs of DTD and ATD.** While performing protein BLAST–based in silico search for DTD sequences, we came across many sequences in the database which are annotated as probable DTD2. However, these sequences bear no sequence similarity to the canonical DTD2 present in Archaea. Therefore, we renamed this protein ATD (as explained later) to distinguish it from the canonical DTD2 and avoid confusion over nomenclature. Moreover, DTD and ATD share < 30% sequence identity between them which is significantly lower than that between DTDs (>50%) or between ATDs (>45%) (Supplementary Fig. 1a). Besides, ATD also does not show homology with DTD3. Multiple sequence alignment of ATD and DTD sequences showed that ATD has −PQATL− and −TNGPYTH− as signature motifs, which are similar to though distinct from the corresponding active site motifs in DTD, viz., −SQFTL− and −NXGPVT−, respectively (Fig. 1a). Strikingly, some of the key conserved residues near DTD's active site, involved in a network of interactions and responsible for holding the Gly-*cis*Pro motif, are also different in ATD. The most notable among these is a highly conserved arginine in DTD (Arg7 in DTD from *Escherichia coli* (EcDTD) or *Plasmodium falciparum* (PfDTD)), which is replaced by a conserved glutamine in ATD (Gln16 in ATD from *Mus musculus* (MmATD)) (Fig. 1a). Thus, comparative analysis of ATD and DTD sequences showed subtle variations in some of the key conserved residues present in and near the active site.

**Phylogenetic distribution of ATD.** A thorough in silico search for ATD sequences revealed that ATD is present in Eukarya, but absent in Bacteria and Archaea. Within Eukarya, ATD is present exclusively in kingdom Animalia, except for a few protozoa (four species of *Leishmania*, two of *Trypanosoma*, and one each of *Saprolegnia*, *Salpingoeca* and *Acanthamoeba*, whose genomes have been sequenced) (Supplementary Data 1). More importantly, ATD is found all across phylum Chordata, whereas its distribution in non-chordate phyla is rather sparse (Supplementary Fig. 1b and Supplementary Data 1). It is worth noting that most of the protozoa that harbor ATD are parasites of various

vertebrate hosts. Therefore, ATDs from these protozoa may be outliers as the possibility of horizontal transfer of ATD gene to these protozoa from their host organisms cannot be ruled out. Contrary to ATD's restricted distribution in Animalia, DTD is found throughout Bacteria and Eukarya. Nevertheless, phylogenetic analysis of ATD and DTD showed that the two fall into two distinct groups (Fig. 1b).

**ATD belongs to the DTD-like fold.** To gain insights into ATD's function, we solved the crystal structure of MmATD at 1.86 Å resolution (Supplementary Fig. 2a and Table 1). We were able to solve the structure by molecular replacement using PfDTD as the search model, despite the fact that the two share < 30% sequence identity. Structural superposition of MmATD on PfDTD and NTD from *Pyrococcus abyssi* (PabNTD) showed an r.m.s.d. of 1.68 Å over 141 Cα atoms and 3.34 Å over 77 Cα atoms, respectively (Fig. 1c and Supplementary Fig. 2b). As is the case with DTD and NTD, ATD too is a homodimeric protein. Interestingly, a Dali-based PDB search for structural homologs of ATD identified a protein (ATD) from *Leishmania major* (LmATD), which is annotated as a probable eukaryotic DTD, and deposited by Structural Genomics of Pathogenic Protozoa Consortium[52]. Structural superimposition of LmATD on MmATD gives an r.m.s.d. of 1.29 Å over 148 Cα atoms (Supplementary Fig. 2c). Thus, like DTD and NTD, ATD also belongs to the DTD-like fold.

**A Gly-*trans*Pro motif in the active site of ATD.** The crystal structure of ATD revealed that its characteristic motifs

| | MmATD |
|---|---|
| **Data collection** | |
| Space group | $P2_12_12_1$ |
| Cell dimensions: | |
| $a$ (Å) | 43.37 |
| $b$ (Å) | 75.35 |
| $c$ (Å) | 103.61 |
| Resolution range (Å)[a] | 50–1.86 (1.93–1.86) |
| Total observations | 166914 |
| Unique reflections | 29268 (1473) |
| Completeness (%) | 89.5 (51.2) |
| $R_{merge}$ (%) | 7.6 (38.0) |
| $<I/(\sigma)I>$ | 21.8 (2.4) |
| Redundancy | 6.4 (3.7) |
| **Data refinement** | |
| Resolution (Å) | 1.86 |
| No. of reflections | 24840 |
| $R_{work}$ (%) | 20.35 |
| $R_{free}$ (%)[b] | 24.96 |
| Monomers/a.u. | 2 |
| No. of residues | 325 |
| No. of atoms | 2721 |
| Protein | 2502 |
| Water | 219 |
| R.m.s. deviation | |
| Bond lengths (Å) | 0.018 |
| Bond angles (º) | 2.040 |
| Mean $B$ value (Å²) | |
| Protein | 31.3 |
| Water | 35.6 |

**Table 1 Crystallographic data collection and refinement statistics**

[a] Values in parentheses are for the highest resolution shell
[b] Throughout the refinement, 5% of the total reflections were held aside for $R_{free}$

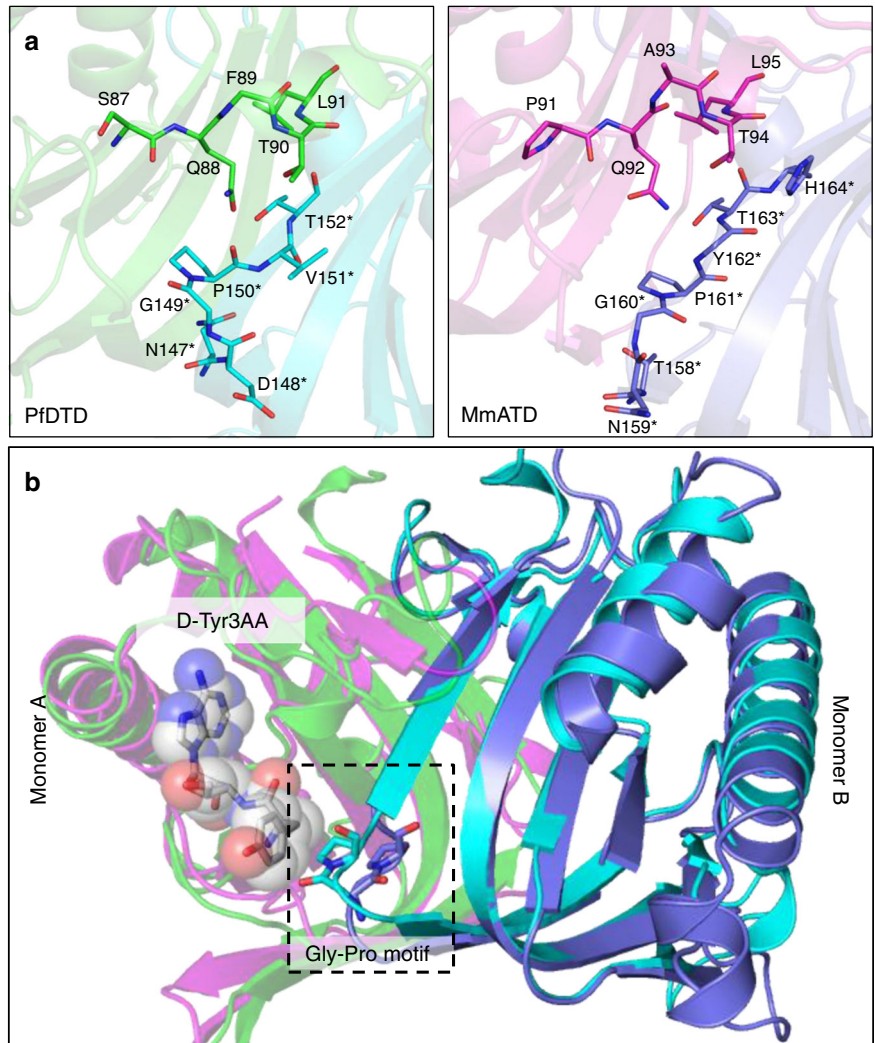

**Fig. 2** ATD has similar active site features as compared to DTD's. **a** Crystal structures of PfDTD (PDB id: 4NBI) and MmATD showing that motifs 1 and 2 form the active site at the dimeric interface in both. **b** Structural superposition of MmATD on PfDTD displaying the cross-subunit Gly-Pro motif in both, i.e., the motif from one monomer inserted into the active site of the other monomer. Residues from the dimeric counterpart are indicated by *

(−PQATL− and −TNGPYTH−) are present at the dimeric interface like the corresponding active site motifs of DTD (−SQFTL− and −NXGPVT−) (Fig. 2a). Like DTD's Gly-*cis*Pro motif, the Gly-*trans*Pro motif of one monomer in ATD is inserted into the active site of the other monomer, i.e., the Gly-Pro motif in DTD as well as ATD is cross-subunit in nature (Fig. 2b and Supplementary Movie 1). Besides the elements of DTD-like fold, specific interactions at the adenine-binding site for the recognition of A76 of tRNA are also highly conserved in ATD (Supplementary Fig. 2d). Surprisingly, MmATD's Gly-Pro motif occurs in *trans* conformation, unlike DTD's Gly-Pro motif which always exists in *cis* conformation as observed in 107 protomers of 19 crystal structures from 5 different organisms (Fig. 3a, b and Supplementary Fig. 3a). Notably, LmATD also possesses a cross-subunit Gly-*trans*Pro motif like MmATD (Supplementary Fig. 3b). Atomic B-factor analysis further revealed that ATD's Gly-*trans*Pro motif is rigid like DTD's Gly-*cis*Pro motif (Supplementary Fig. 4a). In addition, ATD's Gly-Pro residues exhibit a dramatic change of ~180° in ψ torsion angle when compared to DTD's Gly-Pro residues due to remodeling of the local network of interactions in the vicinity of active site (Fig. 3c, d and Supplementary Movie 2). DTD's Gly-*cis*Pro carbonyl oxygens are parallel and protrude into the active site pocket away from the

protein core, i.e., "outward parallel" orientation. ATD's Gly-*trans*Pro carbonyl oxygens are also parallel, but they face away from the active site toward the protein core, i.e., "inward parallel" orientation (Fig. 3a, b). Thus, a direct consequence of *cis*-to-*trans* switch has a marked influence on the orientation of the carbonyl oxygens of glycine and proline residues of the Gly-Pro motif that is responsible for L-chiral rejection in DTD.

**Migration of a conserved arginine.** Upon further analysis of the active site region, it was observed that Arg7 in EcDTD or PfDTD, which is highly conserved in DTDs, is replaced by a conserved glutamine in ATD (Gln16 in MmATD). Interestingly, an invariant arginine is present in a totally different position in ATD (Arg151 in MmATD) (Figs. 1a and 3d). The side chain of Arg7 in PfDTD interacts with the main chain of Gly-*cis*Pro motif from the same monomer, thereby locking the motif rigidly in *cis* conformation (Fig. 3d). This side chain–main chain interaction is conserved in all the available structures (107 protomers) of DTD (Supplementary Fig. 4b). In contrast, the interaction of MmATD's Arg151 side chain with the main chain of Gly-*trans*Pro motif from the dimeric counterpart pulls the motif's backbone outwards, thus holding the motif rigidly in *trans* conformation (Fig. 3d and

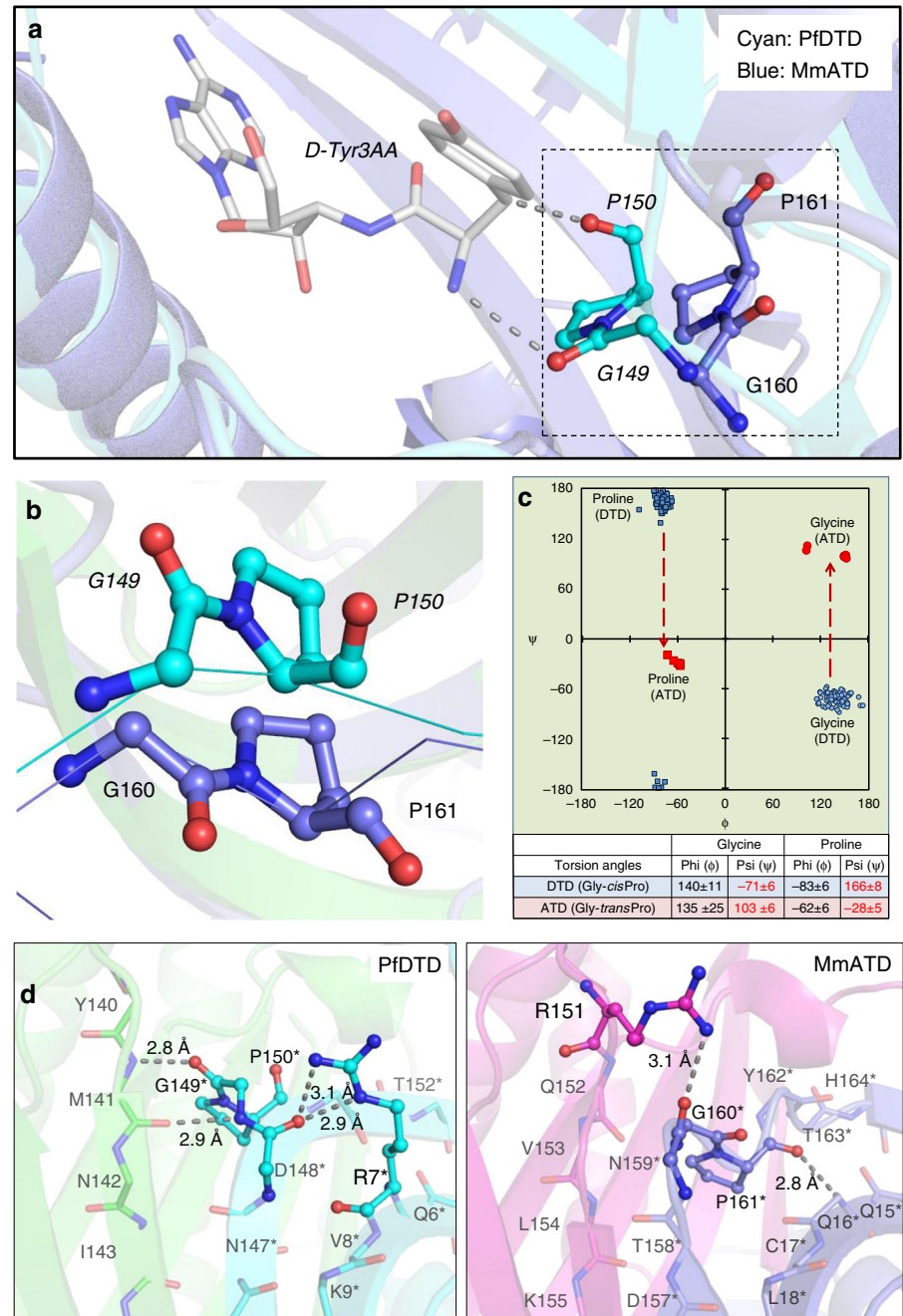

**Fig. 3** ATD has a Gly-*trans*Pro motif in the active site, unlike a Gly-*cis*Pro motif in DTD. **a** Comparison between Gly-*trans*Pro motif in MmATD and Gly-*cis*Pro motif in PfDTD (PDB id: 4NBI) after structural superposition of the two proteins. **b** The comparison shown in **a** depicted from a different angle, highlighting the opposite orientation of Gly-Pro carbonyl oxygens of the two proteins. **c** Ramachandran plot of glycine and proline residues of the Gly-Pro motif of all the available crystal structures of DTD (blue) and ATD (red), highlighting the change of ~180° in the $\psi$ torsion angle. **d** Interaction of the side chain of Arg7 with the Gly-*cis*Pro motif of the same monomer in PfDTD (PDB id: 4NBI), and of the side chain of Arg151 with the Gly-*trans*Pro motif of the dimeric counterpart in MmATD. Residues from the dimeric counterpart are indicated by *

Supplementary Fig. 4b). Hence, the highly conserved arginine in the vicinity of DTD's active site has migrated to a different position near ATD's active site. This suggests that in addition to other elements of the active site, the conserved arginines of DTD and ATD will have a role in the *cis*-to-*trans* switch of the Gly-Pro motif, an aspect which needs further probing.

**"Additional" space in the active site of ATD.** In DTD, the "outward parallel" orientation of Gly-*cis*Pro carbonyl oxygens acts as a "chiral selectivity filter" to strictly reject all L-amino acids

from the pocket through steric exclusion[11,20]. Comparative analysis of active sites of DTD and ATD further revealed that the inward flip of ATD's carbonyl oxygens due to *trans* conformation of its Gly-Pro motif has created "additional" space in its active site when compared to DTD. Consequently, ATD can easily accommodate a larger group in that space as opposed to just hydrogen in DTD. This clearly suggests that ATD can cradle small L-amino acids in its active site pocket (Supplementary Fig. 5, Supplementary Table 1 and Supplementary Movie 1). The "additional" space created as a consequence of

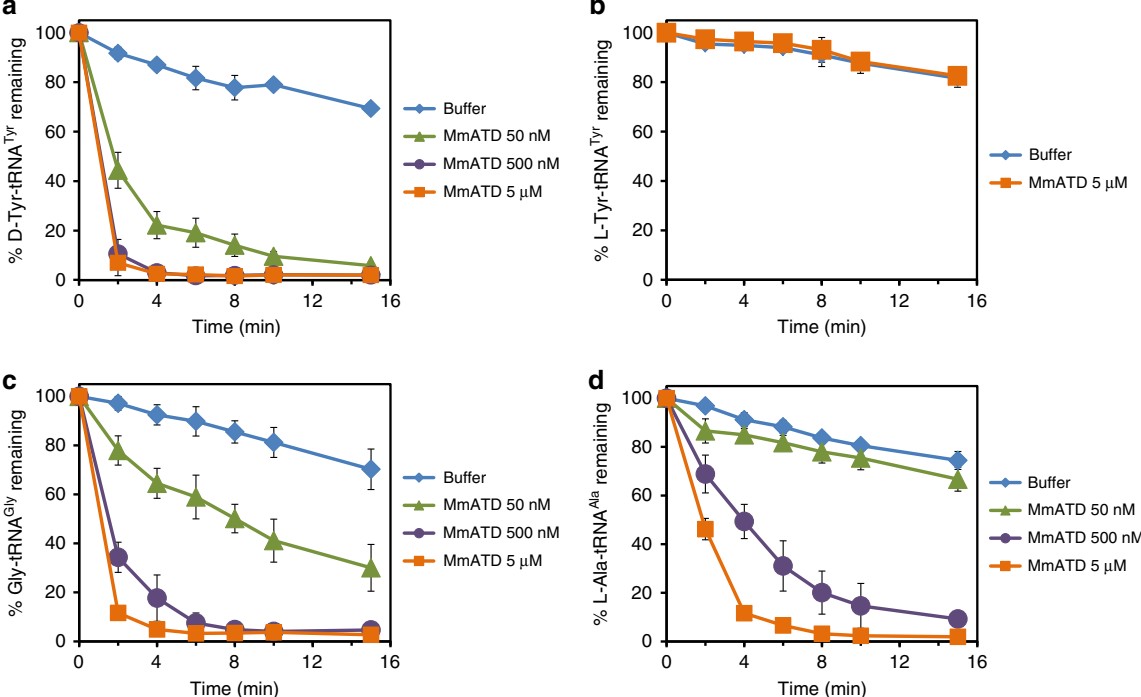

**Fig. 4** ATD displays relaxation of substrate chiral specificity. **a–d** Deacylation of D-Tyr-tRNA^Tyr, L-Tyr-tRNA^Tyr, Gly-tRNA^Gly and L-Ala-tRNA^Ala by different concentrations of MmATD. Aminoacyl-tRNAs were used at 200 nM final concentration. Error bars denote one standard deviation from the mean of three independent readings

Gly-*trans*Pro switch in ATD prompted us to check its biochemical activity profile on L-aminoacyl-tRNAs, in addition to D-aminoacyl- and glycyl-tRNAs.

**Relaxed substrate chiral specificity of ATD**. The fact that ATD belongs to the DTD-like fold and its active site elements and architecture are similar to those of DTD indicated that it could be acting on some non-cognate aminoacyl-tRNA and hence could be involved in translational proofreading. Therefore, we generated the biochemical activity profile of MmATD by screening a spectrum of aminoacyl-tRNAs having either D- or L-amino acid, as well as Gly-tRNA^Gly. MmATD shows significant activity at 50 nM concentration on D-Tyr-tRNA^Tyr, but fails to act on the L-counterpart even at 100-fold higher concentration (Fig. 4a, b and Supplementary Table 2). It also deacylates Gly-tRNA^Gly at 500 nM concentration (Fig. 4c and Supplementary Table 2). Thus, like DTD, ATD acts on both D-chiral and achiral substrates, albeit with significantly less efficiency. Strikingly, when tested with L-Ala-tRNA^Ala, 500 nM MmATD displayed noticeable activity (Fig. 4d and Supplementary Table 2). By contrast, EcDTD or PfDTD does not act on L-chiral substrates even at 100-fold higher concentration than required for D-chiral substrate[11,20]. Therefore, biochemical probing suggested that ATD is an aminoacyl-tRNA deacylase with a relaxed specificity for substrate chirality, primarily due to the *trans* conformation of its active site Gly-Pro motif. It is for this reason that we named this protein ATD, which stands for <u>A</u>nimalia-specific <u>t</u>RNA <u>d</u>eacylase. ATD thus has a "gain of function" in L-chiral activity when compared to DTD due to the switch from Gly-*cis*Pro to Gly-*trans*Pro. Furthermore, biochemical data in conjunction with structural data indicate that ATD's active site pocket, because of the "additional" space created by the inward movement of the Gly-Pro carbonyl oxygens, can accommodate only small L-amino acids like L-alanine but not the bulkier ones like L-tyrosine.

**Proofreading of L-Ala-tRNA^Thr(G4•U69) by ATD**. While we were in the process of identifying the physiological substrate for ATD, a recent study reported that eukaryotic AlaRS has acquired the function of L-alanine mischarging on multiple non-cognate tRNAs harboring G4•U69 wobble base pair in the acceptor stem. One of the tRNAs that undergo significant levels of such mischarging is tRNA^Thr(G4•U69)[53]. This is in addition to the canonical recognition of AlaRS-specific universally occurring G3•U70 in tRNA^Ala[54,55]. Interestingly, such an error in the selection of tRNAs bearing G4•U69 was found only in the case of eukaryotic AlaRS and not the bacterial one[53]. In this context, therefore, eukaryotic AlaRS can be called non-discriminating AlaRS (AlaRS^ND), whereas bacterial AlaRS can be referred to as discriminating AlaRS (AlaRS^D). The above finding prompted us to test the role of ATD in proofreading tRNA^Thr(G4•U69) selection error made by eukaryotic AlaRS^ND. Strikingly, biochemical assays with MmATD showed significantly higher selectivity for the non-cognate L-Ala-tRNA^Thr(G4•U69) as the enzyme acts on the substrate at just 1 nM compared to its activity on the cognate L-Thr-tRNA^Thr(G4•U69) at 50 nM (Fig. 5a and Supplementary Table 2). Thus, MmATD displays at least 50-fold difference in biochemical activity on these two substrates, indicating that L-Ala-tRNA^Thr(G4•U69) is the preferred substrate for ATD.

The other non-cognate tRNA that was found to be significantly mischarged by eukaryotic AlaRS was tRNA^Cys(G4•U69)[53]. We therefore checked ATD's biochemical activity on L-Ala-tRNA^Cys(G4•U69) to test its role in clearing the misacylated species. It was observed that MmATD acts on the substrate at 50 nM concentration, indicating that L-Ala-tRNA^Cys(G4•U69) may not be the physiological substrate (Supplementary Fig. 6 and Supplementary Table 2). Thus, a comparison of biochemical activities of MmATD on different aminoacyl-tRNA substrates suggests that L-Ala-tRNA^Thr(G4•U69) is the principal substrate of ATD, whereas L-Ala-tRNA^Cys(G4•U69) may be partly cleared in

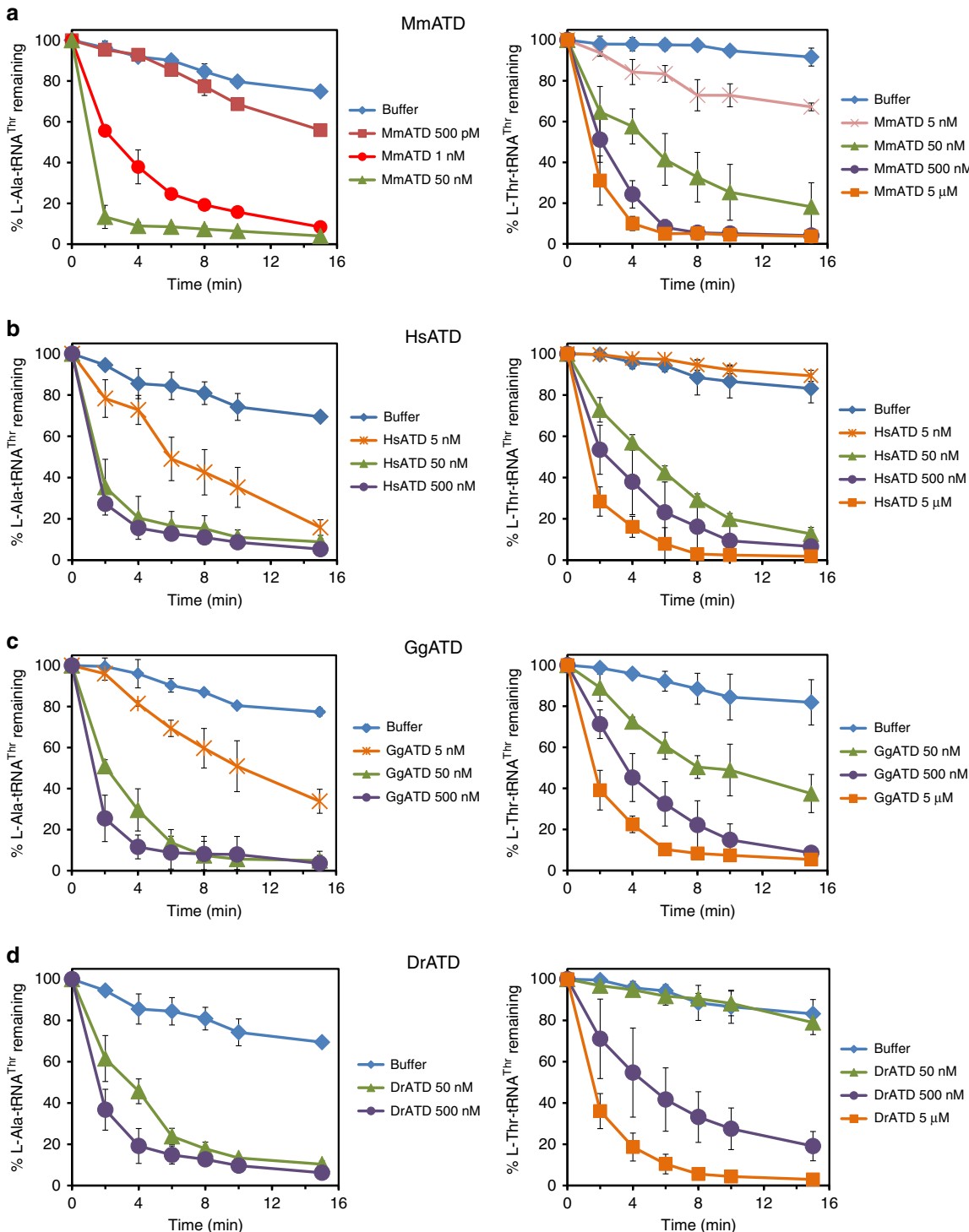

**Fig. 5** Proofreading of L-Ala-tRNA$^{Thr}$(G4•U69) by ATD is conserved across organisms. Deacylation of L-Ala-tRNA$^{Thr}$(G4•U69) and L-Thr-tRNA$^{Thr}$ (G4•U69) by different concentrations of **a** MmATD, **b** HsATD, **c** GgATD, and **d** DrATD. Aminoacyl-tRNAs were used at 200 nM final concentration. Er`ror bars denote one standard deviation from the mean of three independent readings

the cellular context. The latter argument is supported by the observation that in the proteomics study of HEK293T cells, misincorporation of L-alanine was observed at cysteine positions but not at threonine positions even though significant misacylation of tRNA$^{Thr}$(G4•U69) with alanine was observed in ex vivo tRNA microarray experiments[53]. Nevertheless, this observation is striking, since in humans, the enrichment of G4•U69 is

significantly more in tRNA$^{Thr}$ genes (20%) than in tRNA$^{Cys}$ genes (3.4%) (Supplementary Table 3).

To rule out the role of DTD in proofreading L-Ala-tRNA$^{Thr}$ (G4•U69), we tested the biochemical activity of DTD from *M. musculus* (MmDTD) on L-Ala-tRNA$^{Thr}$(G4•U69). It was observed that MmDTD does not act on the substrate at even 100 nM concentration (Supplementary Fig. 7), which is in stark

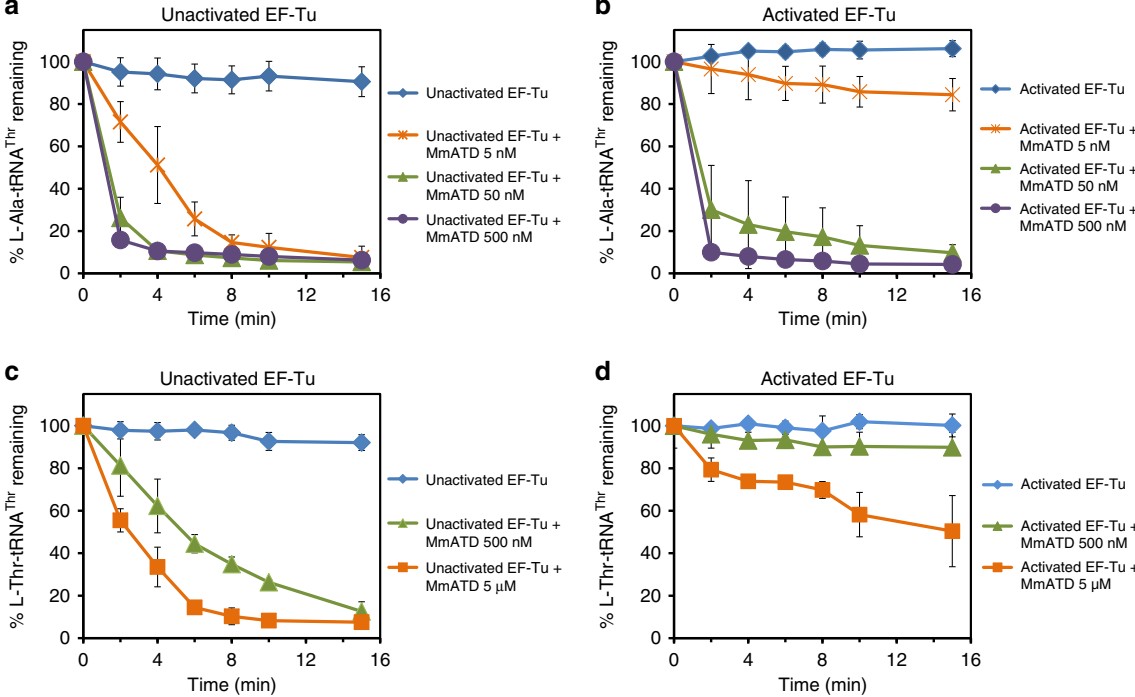

**Fig. 6** EF-Tu confers protection on L-Thr-tRNA$^{Thr}$(G4•U69) against ATD. Deacylation of L-Ala-tRNA$^{Thr}$(G4•U69) by different concentrations of MmATD in the presence of **a** unactivated EF-Tu, and **b** activated EF-Tu. Deacylation of L-Thr-tRNA$^{Thr}$(G4•U69) by different concentrations of MmATD in the presence of **c** unactivated EF-Tu, and **d** activated EF-Tu. Aminoacyl-tRNAs and EF-Tu were used at 200 nM and 2 μM final concentration, respectively. Error bars denote one standard deviation from the mean of three independent readings

contrast to MmATD's activity on the same substrate at just 1 nM (Fig. 5a). DTD's inability to act on the substrate is expected, since the chiral proofreading enzyme functions through strict L-chiral rejection to achieve its remarkable enantioselectivity[11,20].

**Conservation of ATD's biochemical activity**. To rule out any organism-specific phenomenon regarding ATD's biochemical activity, we tested ATDs from multiple organisms belonging to different taxonomic groups under Chordata—human (*Homo sapiens*, HsATD) of class Mammalia (mammals), chicken (*Gallus gallus*, GgATD) of class Aves (birds), and zebrafish (*Danio rerio*, DrATD) of class Pisces (fishes). It was observed that all these ATDs can act on L-Ala-tRNA$^{Thr}$(G4•U69) more efficiently than on L-Thr-tRNA$^{Thr}$(G4•U69) (Fig. 5b–d). Therefore, not only ATD's activity on the non-cognate substrate but also its ability to discriminate between L-Ala-tRNA$^{Thr}$(G4•U69) and L-Thr-tRNA$^{Thr}$(G4•U69) is conserved across diverse classes of Chordata. We then analyzed tRNA$^{Thr}$(G4•U69) gene sequences from diverse organisms which revealed that the first five base pairs in the acceptor stem are invariant or highly conserved (Supplementary Fig. 8). As ATD belongs to the DTD-like fold, its interaction with the tRNA is not expected to go beyond the first three or four base pairs in the acceptor stem. Hence, lack of variation in the acceptor stem of tRNA$^{Thr}$(G4•U69) further suggests that ATD's biochemical activity on L-Ala-tRNA$^{Thr}$(G4•U69) is conserved across diverse taxonomic groups.

**Protection of L-Thr-tRNA$^{Thr}$(G4•U69) from ATD by EF-Tu**. Since ATD had shown significant activity on L-Thr-tRNA$^{Thr}$(G4•U69), we checked whether EF-Tu can protect the cognate substrate from ATD. EF-Tu occurs abundantly in the cell[56] and most aminoacyl-tRNAs are expected to exist in complex with EF-Tu, ready for delivery to the ribosome. On the basis of

thermodynamic compensation, EF-Tu is expected to bind the non-cognate L-Ala-tRNA$^{Thr}$(G4•U69) with significantly lower affinity compared to the cognate L-Thr-tRNA$^{Thr}$(G4•U69)[57]. Competition assays demonstrated that L-Ala-tRNA$^{Thr}$(G4•U69) undergoes significant deacylation with 50 nM MmATD in the presence of EF-Tu (Fig. 6a, b). By contrast, L-Thr-tRNA$^{Thr}$(G4•U69) is completely protected by EF-Tu even at 500 nM enzyme, whereas its protection is marginally relieved at 5 μM MmATD (Fig. 6c, d). Hence, the discrimination potential/factor of MmATD for these two substrates enhances to more than 100-fold in the presence of EF-Tu (Figs. 5a and 6b, d). The above biochemical data clearly suggest that L-Ala-tRNA$^{Thr}$(G4•U69) is ATD's physiological substrate, and EF-Tu is able to confer adequate protection on the cognate substrate against ATD.

**Correlation between tRNA$^{Thr}$(G4•U69) mischarging and ATD**. To ascertain whether any correlation exists between ATD and tRNA$^{Thr}$(G4•U69), we performed a thorough bioinformatic analysis. It showed that many Animalia genomes are enriched in G4•U69-containing tRNA genes of which tRNA$^{Thr}$(G4•U69) genes exhibit the highest level of enrichment. The enrichment of tRNA$^{Thr}$(G4•U69) genes ranges from 20 to 40% (average ~30%). Such an enrichment of tRNA$^{Thr}$(G4•U69) genes is observed throughout phylum Chordata, as well as in one organism (*Strongylocentrotus purpuratus*) from phylum Echinodermata whose tRNA gene sequences are available in the database. The enrichment of G4•U69 is markedly less in other tRNA genes compared to tRNA$^{Thr}$ genes. For example, tRNA$^{Cys}$(G4•U69) genes constitute only 0.69–11.9% (average ~5%) of total tRNA$^{Cys}$ genes (Fig. 7a, b, Supplementary Table 3 and Supplementary Data 1). In addition, among all the G4•U69-containing tRNA genes found in Chordata, only tRNA$^{Thr}$(G4•U69) genes occur in all those chordate species whose tRNA gene sequences are available in the database. Other tRNA genes carrying G4•U69 are

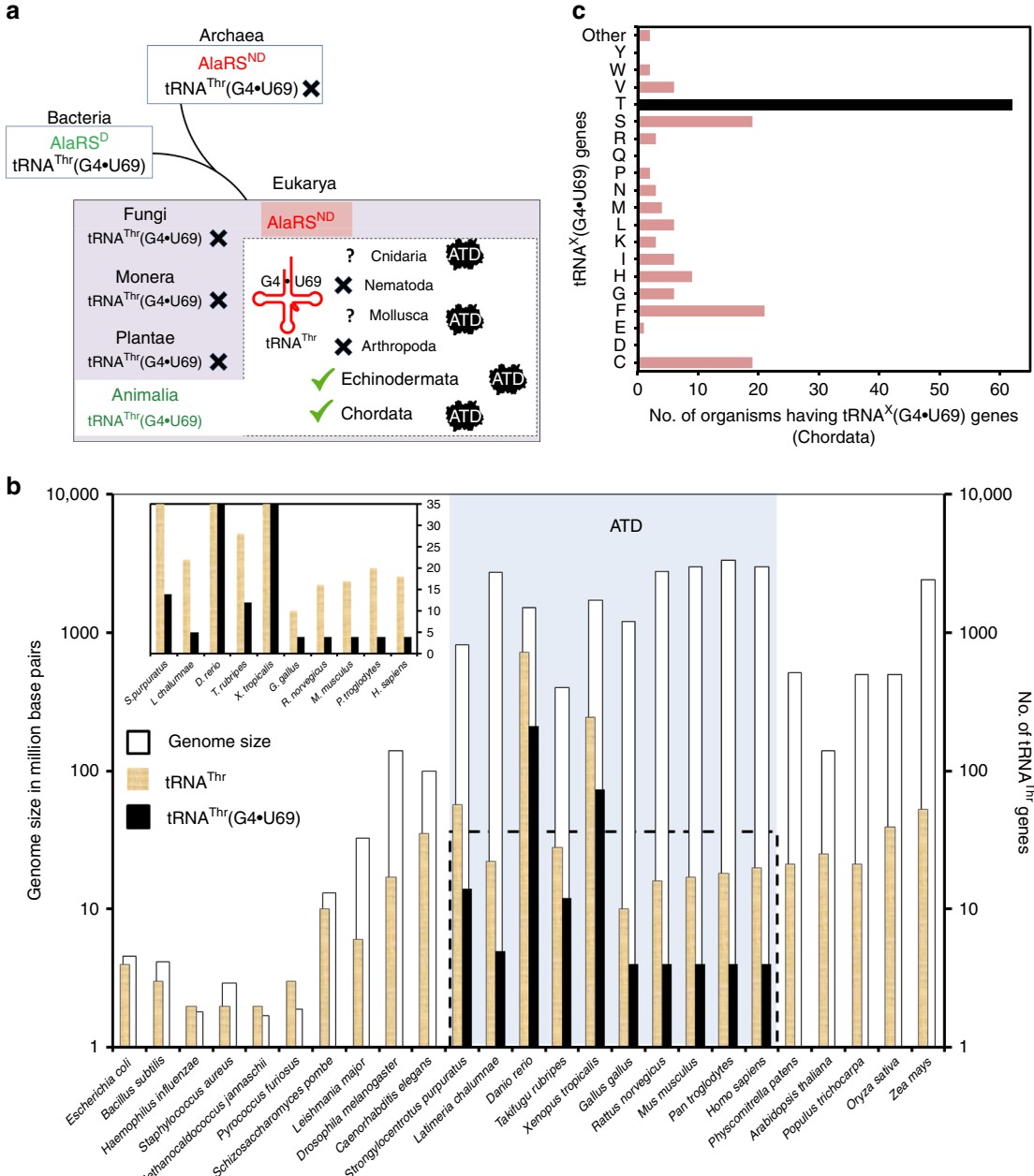

**Fig. 7** Enrichment of tRNA<sup>Thr</sup>(G4•U69) genes and presence of ATD show strict association. **a** Distribution of AlaRS<sup>ND</sup>, tRNA<sup>Thr</sup>(G4•U69) genes, and ATD in different domains of life. tRNA gene sequences of Cnidaria and Mollusca are not available in the database. **b** Bar graph (logarithmic scale) depicting genome size, total number of tRNA<sup>Thr</sup> genes, and number of tRNA<sup>Thr</sup>(G4•U69) genes occurring in representative organisms belonging to all the three domains of life. Inset showing the number of total tRNA<sup>Thr</sup> genes and tRNA<sup>Thr</sup>(G4•U69) genes in normal scale; genome size has not been shown for the sake of clarity. Presence of ATD is highlighted in light blue box. Data for occurrence of AlaRS<sup>ND</sup> and tRNA<sup>Thr</sup>(G4•U69) genes have been taken from reference [53]. **c** Bar graph showing the number of organisms containing (G4•U69)-harboring tRNA genes which code for tRNAs specific for various proteinogenic amino acids

restricted to only a small subset of organisms. For instance, tRNA<sup>Cys</sup>(G4•U69) genes occur in only 19 of 62 chordate species whose tRNA gene sequences are available in the database (Fig. 7c). The above observations further indicate that L-Ala-tRNA<sup>Thr</sup>(G4•U69) constitutes the major substrate of ATD, whereas others including L-Ala-tRNA<sup>Cys</sup>(G4•U69) are only minor ones.

Strikingly, a survey for the presence of ATD revealed its strict association with the enrichment of tRNA<sup>Thr</sup>(G4•U69) genes (Fig. 7a, b). Remarkably, organisms (e.g., *Drosophila melanogaster*) that lack tRNA<sup>Thr</sup>(G4•U69) genes simultaneously lack ATD,

including archaea which also seem to possess eukaryotic-type AlaRS<sup>ND</sup>. Although many bacteria possess tRNA<sup>Thr</sup>(G4•U69) genes, they lack AlaRS<sup>ND</sup> altogether and hence, the problem of L-alanine misacylation of tRNA<sup>Thr</sup>(G4•U69) does not arise at all. Thus, the problem of mischarging of tRNA<sup>Thr</sup>(G4•U69) with L-alanine arises only, when tRNA<sup>Thr</sup>(G4•U69) is present alongside AlaRS<sup>ND</sup>. Such a strong as well as strict correlation between ATD and the problem of tRNA<sup>Thr</sup>(G4•U69) mis-selection by AlaRS<sup>ND</sup>, in terms of either concomitant occurrence or concomitant absence, clearly points toward a functional link between ATD and proofreading of error in tRNA<sup>Thr</sup>(G4•U69) selection.

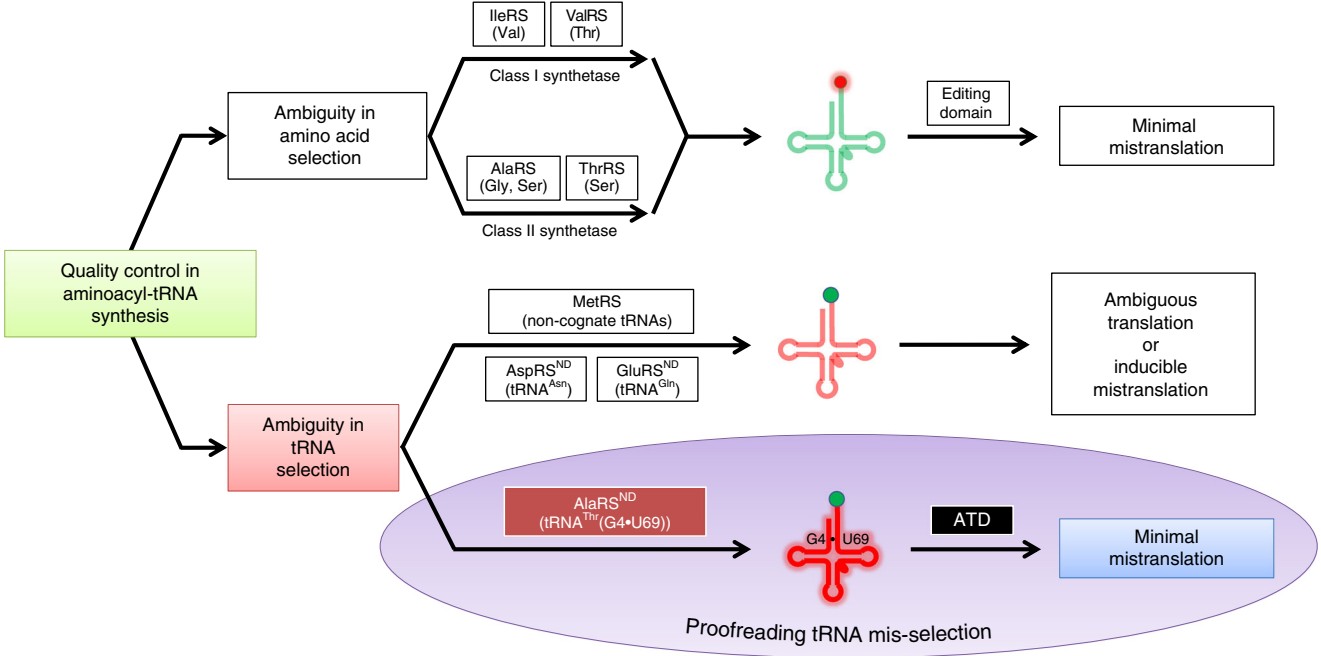

**Fig. 8** ATD is a unique and dedicated proofreading factor that rectifies a critical tRNA selection error. Model for mis-selection and consequent misacylation of tRNA$^{Thr}$(G4•U69) with L-alanine by AlaRS$^{ND}$, and its subsequent proofreading by ATD. Cognate and non-cognate tRNAs (clover leaf model) are colored in green and red, respectively. Likewise, cognate and non-cognate amino acids (circle) are rendered in green and red, respectively

## Discussion

Overall, our extensive structural and biochemical probing in combination with in-depth in silico analysis confirms that ATD serves as a dedicated factor for correcting the tRNA$^{Thr}$(G4•U69) selection error committed by eukaryotic AlaRS$^{ND}$ (Fig. 8). ATD thus rectifies a critical tRNA mis-selection rather than a mistake in amino acid selection by a synthetase that has been extensively studied so far in the context of proofreading[28–38]. Such an error correction capability has not been attributed to any of the known editing domains, although ambiguous tRNA selection happens in several instances, wherein the ambiguity imparts selective advantage to the system[7,58] (Fig. 8). Besides, it also suggests that the evolutionary gain of function by AlaRS$^{ND}$ in charging G4•U69-bearing tRNAs with L-alanine may be advantageous, but may also require factors like ATD to keep such "errors" below precarious levels, thus avoiding global misfolding and cell death.

The study also highlights how a *cis*-to-*trans* switch in the active site of DTD-like fold has led to a dramatic change in substrate chiral specificity of ATD when compared to DTD. An invariant cross-subunit Gly-*cis*Pro dipeptide motif in the active site of DTD is crucial for the enzyme's enantioselectivity[11]. The *cis* conformation of the motif in DTD disposes the Gly-Pro carbonyl oxygens in a near-parallel orientation, projecting them directly into the active site pocket. This results in strict rejection of even L-alanine (the smallest L-amino acid) from DTD's chiral proofreading site. Thus, DTD functions only through L-chiral rejection rather than D-chiral selection, and hence acts on D-amino acids, as well as achiral glycine. The porosity of DTD's active site to glycine leads to Gly-tRNA$^{Gly}$ "misediting paradox", which is resolved through protection of the cognate achiral substrate by EF-Tu[20]. We have recently elucidated the functional importance of DTD's activity on glycine as DTD acts an efficient editor of the mischarged Gly-tRNA$^{Ala}$ species generated by AlaRS[59]. In contrast to DTD, ATD harbors an invariant cross-subunit Gly-*trans*Pro dipeptide motif in its active site. The *trans* conformation of the motif in ATD orients the Gly-Pro carbonyl oxygens away

from the active site pocket, i.e., about 180° flip with respect to DTD's Gly-Pro carbonyl oxygens. This creates "additional space" in ATD's active site pocket when compared to that of DTD (Supplementary Fig. 5), thereby causing relaxation in the stereospecificity of ATD. The latter therefore displays a gain of function as it is able to act on L-chiral substrates, in addition to D-chiral and achiral substrates. However, differences observed in the activity of ATD on various substrates suggest that tRNA plays a significant role in modulating ATD's activity, similar to what appears to be true in the case of DTD.

The role of ATD in Animalia to specifically prevent, minimize or regulate substitution of L-alanine for L-threonine in proteins may be crucial. It is tempting to speculate that threonine-to-alanine mutations will modulate a diverse array of phosphorylation sites on proteins, thereby causing a drastic modification of the cellular phosphoproteome. The regulatory function, if any, of ATD in generating such proteome diversity, thereby providing selective advantage to a cell or tissue type and under specific conditions such as pathogenesis or immune response, needs to be explored through up-/down-regulation, as well as by using knockout approaches in multiple systems. The genome size and the number of tRNA genes both exhibit a drastic increase in going from prokaryotes to eukaryotes. However, in higher eukaryotes such as mammals, the number of tRNA isodecoders shows a progressive increase with the increase in genome size, although the total number of tRNA genes shows negligible change. This implies that with increasing genome size in eukaryotes there is a significant enhancement in tRNA sequence variation[60]. It has been recently suggested that the size of tRNA limits the identity determinants required for faithful translation without cross-reacting with non-cognate synthetases[61]. As has been noted in the current work, the expansion of genome size (from around 100 million base pairs in non-Chordata to >1 billion base pairs in Chordata) has led to such a cross-reactivity and enhancement in tRNA mis-selection, thereby necessitating recruitment of dedicated factors for error correction (Fig. 7b). Identification of ATD

in the present study provides a remarkable example of such a scenario. The advent of ATD thus marks a key event associated with the appearance of Animalia, and more specifically of Chordata about 500 million years ago, to ensure translational quality control.

## Methods

**Protein expression and purification**. The genes encoding ATDs, *M. musculus* DTD (residues 1–147) and *M. musculus* ThrRS(ΔNTD) (residues 320–721) were PCR-amplified from respective cDNAs and inserted into pET-28b vector using restriction-free cloning[62]. To generate C-terminal 6× His-tagged protein, the stop codon was removed from the reverse primer. *M. musculus* full-length AlaRS gene was PCR-amplified from mouse cDNA and inserted into pET-28b vector between *Nde*I and *Xho*I sites using conventional restriction-based cloning. The recombinant proteins were overexpressed in Rosetta$^{TM}$(DE3) strain of *E. coli*. Purification of His-tagged proteins (ATDs, DTD, AlaRS and ThrRS) was done using a two-step protocol, i.e., Ni-NTA affinity chromatography followed by size exclusion chromatography (SEC). The storage buffer for ATDs and DTD contained 100 mM Tris (pH 8.0) and 200 mM NaCl, while that for AlaRS and ThrRS comprised 50 mM Tris (pH 8.0), 150 mM NaCl and 5 mM 2-mercaptoethanol (β-ME). Un-tagged MmATD was purified using cation exchange chromatography (CEC) followed by SEC. For CEC, ATD-overexpressing *E. coli* cell pellet was lysed in a buffer containing 50 mM Bis–Tris (pH 6.5) and 20 mM NaCl, and the supernatant was subjected to chromatographic separation using sulfopropyl sepharose (GE Healthcare Life Sciences, USA). The protein of interest was eluted using a NaCl gradient of 20–500 mM. For SEC, Superdex 75 column (GE Healthcare Life Sciences, USA) was used.

**Crystallization of MmATD**. The purified un-tagged MmATD was screened for crystallization conditions using different screens (Index, Crystal Screen HT, PEG/ Ion and PEGRx from Hampton Research, USA) at two different temperatures—4 ° C and 20 °C. Mosquito Crystal (TTP LabTech, UK) crystallization robot was used to set up crystallization experiments using sitting-drop vapor diffusion method by mixing 1 μl protein and 1 μl reservoir buffer in a 96-well MRC plate with three sub-wells (Molecular Dimensions, UK). The initial hits from the screens were further expanded for optimization using sitting-drop vapor diffusion method in 96-well format MRC plates having three sub-wells. Reservoir buffer with 0.1 M Bicine (pH 8.0) and 15% PEG1500 yielded good diffraction-quality crystals.

**X-ray crystallography**. In-house X-ray facility consisting of RigakuMicromax007 HF with rotating-anode generator and MAR345-dtb image plate detector was used for crystal screening and data collection at 100 K using an Oxford Cryostreamcooler (Oxford Cryosystems, UK). The wavelength of X-rays used was 1.5418 Å, corresponding to Cu Kα radiation. HKL2000[63] was used for data processing and MOLREP-AUTO MR from CCP4 suite[64] for molecular replacement. PfDTD (PDB id: 4NBI), with the ligand removed, was used as the search model for molecular replacement. Refinement and model building were done using REFMAC[65] and COOT[66], respectively. Structure validation was done using PROCHECK[67]. PyMOL Molecular Graphics System, Version 1.7.6.0 Schrödinger, LLC was used to generate figures. Structure-based multiple sequence alignment was carried out using the T-Coffee server in Expresso mode (http://tcoffee.crg.cat/apps/tcoffee/do:expresso), and the corresponding figure was generated using ESPript 3.0 (http://espript.ibcp. fr/ESPript/cgi-bin/ESPript.cgi).

**In vitro biochemical experiments**. tRNAs (*M. musculus* tRNA$^{Gly}$, tRNA$^{Ala}$, tRNA$^{Thr}$(G4•U69), and *E. coli* tRNA$^{Tyr}$) were generated by in vitro transcription of the corresponding tRNA genes using MEGAshortscript T7 Transcription Kit (Thermo Fisher Scientific, USA). tRNAs were end-labeled with [α-$^{32}$P] ATP (BRIT-Jonaki, India) using CCA-adding enzyme[68]. Glycylation of tRNA$^{Gly}$ was done by incubating 1 μM tRNA$^{Gly}$ with 2 μM *Thermus thermophilus* GlyRS in a buffer containing 100 mM HEPES (pH 7.5), 10 mM KCl, 30 mM MgCl$_2$, 50 mM glycine, and 2 mM ATP at 37 °C for 15 min. Alanylation of tRNA$^{Ala}$ and tRNA$^{Thr}$ (G4•U69) was performed by incubating 1 μM tRNA$^{Ala}$ and 10 μM tRNA$^{Thr}$ (G4•U69) with 8 μM *M. musculus* AlaRS in a solution composed of 100 mM HEPES (pH 7.5), 30 mM KCl, 100 mM MgCl$_2$, 10 mM ATP, 10 mM dithiothreitol (DTT), 1 unit/ml of PPase enzyme (Thermo Fisher Scientific, USA) and 10 mM L-alanine at 37 °C for 15 min. Threonylation of tRNA$^{Thr}$(G4•U69) was carried out by incubating 1 μM tRNA$^{Thr}$(G4•U69) and *M. musculus* ThrRS(ΔNTD) in a buffer comprising 100 mM HEPES (pH 7.5), 100 mM MgCl$_2$, 300 mM KCl, 45 mM L-threonine, 2.5 mM DTT and 2 mM ATP at 37 °C for 15 min. tRNA$^{Tyr}$ was aminoacylated by *E. coli* TyrRS, as described[11]. *T. thermophilus* EF-Tu activation and protection assays were done using the protocol, as described[20]. Deacylation assays[11] and EF-Tu rescue experiments[20] were carried out in conditions as described. $k_{obs}$ values were calculated by fitting the data points on the curve according to the first-order exponential decay equation $[S_t] = [S_0]e^{-kt}$, where $[S_t]$ is the substrate concentration at time $t = t$, $[S_0]$ is the substrate concentration at time $t = 0$, and $k$ is the first-order decay constant, i.e., $k_{obs}$; curve fitting and $k_{obs}$ calculations were done using GraphPad Prism software. All the experiments were performed in triplicates, and the mean values were used to plot the graphs. Error bars denote one standard deviation from the mean of three independent readings. All biochemical data sets with corresponding mean values and standard deviations have been provided, as Supplementary Data 2.

**Bioinformatic analysis**. Protein sequences were retrieved from NCBI and were subjected to phylogenetic tree construction using the web server http://www. phylogeny.fr/ (bootstrap number = 100) and iTOL web server http://itol.embl.de/. tRNA gene sequences were retrieved from GtRNAdb and sequences having tRNAscan-SE score > 50 were used for analysis (http://gtrnadb.ucsc.edu/). The list of organisms whose genomes have been completely sequenced was obtained from KEGG GENOME database (http://www.genome.jp/kegg/genome.html). Information about genome size was taken from the web server http://www.bionumbers. hms.harvard.edu/default.aspx. Multiple sequence alignment of tRNA$^{Thr}$ and tRNA$^{Thr}$(G4•U69) was prepared using T-Coffee server in M-Coffee mode (http:// tcoffee.crg.cat/apps/tcoffee/do:mcoffee), while consensus sequence logo was prepared using WebLogo server (http://weblogo.berkeley.edu/logo.cgi).

**Movie preparation**. The two conformations/states, one of PfDTD (initial state; PDB id: 4NBI) and the other of MmATD, were morphed using UCSF Chimera software[69]. Movie was then prepared using PyMOL Molecular Graphics System, Version 1.7.6.0 Schrödinger, LLC.

**Data availability**. The atomic coordinates and structure factors for MmATD have been deposited in the Protein Data Bank with the accession code PDB id 5XAQ. All other data are available from the corresponding author upon reasonable request.

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

## Acknowledgements

We would like to acknowledge Dr. P. Chandra Shekar for kindly providing mouse cDNA. S.K.K. thanks DST-INSPIRE, India, for research fellowship. M.M. thanks Department of Biotechnology, India, for research fellowship. S.B.R. and R.S. acknowledge funding from 12th Five Year Plan Project BSC0113 of CSIR, India. R.S. also acknowledges funding from J.C. Bose Fellowship of SERB, India, and Centre of Excellence Project of Department of Biotechnology, India.

## Author contributions

S.K.K., M.M., R.S., and B.K. designed and performed the experiments. R.S. conceived and supervised the study. All the authors analyzed the data. S.B.R. and R.S. wrote the manuscript with help from S.K.K. and M.M., and all the authors reviewed it.

## Additional information

**Competing interests:** The authors declare no competing financial interests.

