## [Peer Review File · Nature Communications]

PEER REVIEW FILE

Reviewers' comments:

Reviewer #1 (Remarks to the Author):

In this manuscript, Sankaranarayanan and coworkers demonstrate the role of a newly identified Animalia-specific tRNA deacylase (ATD) in editing of mis-acylated Ala-tRNA(Thr).

The authors identify an animal-specific paralog of D-aminoacyl-tRNA deacylase (DTD) with distinct sequence variations near the known active site. Their 1.86 Å structure of the mouse enzyme reveals a DTD-like fold with characteristic dimer interface active site. The presence of a Gly-trans-Pro motif was surprising, given the conserved Gly-cis-Pro motif of DTD enzymes. The authors clearly demonstrate that this more open active site changes the substrate specificity from D-amino acyl-tRNAs to L-Ala-tRNA. Finally, the authors connect this structural determination with the recently identified mis-aminoacylation of tRNAThr(G4-U69) by eukaryotic AlaRS to show that ATD is able to perform trans editing of mis-acylated L-Ala-tRNAThr and to a lesser degree of L-Ala-tRNACys.

This work is novel, clear, and convincing, and it was a pleasure to read. The supplemental movie files were useful. The authors have carried out a comprehensive in silico analysis of the distribution of ATD in addition to their structural and kinetic work. The selective co-enrichment of ATD in species containing abundant tRNA(Thr) G4:U79 is compelling.

Minor points:

The authors claim that the error in amino acid selection by aminoacyl-tRNA synthetases is as high as one in 100-1000, which is reasonable based simply on binding affinity. Yet this doesn't take into account the editing capabilities of the synthetases, as the authors certainly know (and with this work contribute to). Perhaps a more full statement would be that because of the low selectivity, these editing functionalities have evolved (primarily in synthetases and associated trans-acting factors) to achieve the ~100-fold higher accuracy in translation.

Style of in-text citations needs to be standardized.

Rebecca Alexander

Reviewer #2 (Remarks to the Author):

The manuscript by Kuncha et al. reports structural and biochemical characterization of a new group of enzymes involved in maintaining fidelity during proteins synthesis. These enzymes, named ATD, are mostly found in Animalia and share sequence and structural similarity with DTD enzymes that hydrolyze D-aminoacyl-tRNAs. Crystal structures clearly show that a key Gly-Pro dipeptide undergoes a cis-to-trans switch from DTD to ATD, which enlarges the active site and allows recognition and hydrolysis of L-aminoacyl-tRNA by ATD. The authors further show that ATD proofreads L-Ala-tRNAThr misformed by AlaRS, providing a previously unknown mechanism of quality control in protein synthesis. Overall, this work is beautifully done and is suitable for the broad readership of Nature Communications. The conclusions, with a minor exception, are well-supported by the results.

Specific comments:

- Comparing structures of ATD and DTD reveals that the position of a key Arg locks Gly-Pro in trans and cis positions, respectively. Has any mutational studies been performed on the Arg residue? Does removing Arg7 allow DTD to hydrolyze L-aa-tRNA, or restoring Arg at the corresponding position abolish ATD's activity towards L-aa-tRNA?
- The concentrations of aa-tRNAs should be included in the legends of Figures 4 and 5.
- The authors should be able to calculate the apparent deacylation rates from Figures 4 and 5. These rates will be useful to better compare substrate selection by ATD.
- Figures 4 and 5, the apparent deacylation rate of MmATD is ~500-fold higher for L-Ala-tRNAThr than for L-Ala-tRNAAla. Is there any structural model to explain how it recognizes the first few pairs of the tRNAThr?
- Has it been tested to see if L-Ala-tRNAThr is a substrate for DTD?
- Line 230 and 640, it's too strong a statement to claim that EF-Tu does not protect L-Ala-tRNAThr from ATD. In fact, Figures 6A and 6B shows that at 5 nM ATD, EF-Tu shows some protective effect. The authors need to revise the statement and related discussion.

Reviewer #3 (Remarks to the Author):

None

Response to reviewers' concerns/comments

Reviewer 1

1. *The authors claim that the error in amino acid selection by aminoacyl-tRNA synthetases is as high as one in 100-1000, which is reasonable based simply on binding affinity. Yet this doesn't take into account the editing capabilities of the synthetases, as the authors certainly know (and with this work contribute to). Perhaps a more full statement would be that because of the low selectivity, these editing functionalities have evolved (primarily in synthetases and associated trans-acting factors) to achieve the ~100-fold higher accuracy in translation.*

Reply:

As suggested by the reviewer, we have now made the necessary change in the revised manuscript.

2. *Style of in-text citations needs to be standardized.*

Reply:

The style of in-text citations has now been standardized in the revised manuscript and conforms to the journal format/guidelines.

Reviewer 2

1. Comparing structures of ATD and DTD reveals that the position of a key Arg locks Gly-Pro in trans and cis positions, respectively. Has any mutational studies been performed on the Arg residue? Does removing Arg7 allow DTD to hydrolyze L-aa-tRNA, or restoring Arg at the corresponding position abolish ATD's activity towards L-aa-tRNA?

Reply:

We thank the reviewer for bringing forth this key aspect about ATD and DTD.

As is discernible in Fig. 1b, there is an intricate network of first- and second-shell interactions, involving both side chains and the main chain, which plausibly holds the Gly-Pro motif in DTD and ATD in its respective conformation. Thus, an extensive and thorough probing via multiple mutations on both DTD and ATD is required to determine the role/contribution of these residues/interactions in maintaining the Gly-Pro motif in DTD and ATD, and this forms a part of our future course of investigation.

Fig. 1(a) Deacylation of L-Ala-tRNA^{Thr}(G4•U69) by MmATD wild-type and Q16R. **(b)** Network of first- and second-shell interactions important for maintaining the Gly-Pro motif conformation in PfDTD and MmATD.

2. *The concentrations of aa-tRNAs should be included in the legends of Figures 4 and 5.*

Reply:

The concentrations of aa-tRNAs and EF-Tu have now been mentioned in the legends of respective Main and Supplementary Figures.

3. *The authors should be able to calculate the apparent deacylation rates from Figures 4 and 5. These rates will be useful to better compare substrate selection by ATD.*

Reply:

We agree with the reviewer that values of apparent deacylation rates provide a better appreciation and comprehension of an enzyme's selectivity towards multiple substrates, and we thank the reviewer for pointing out this useful fact. The following table provides these values, which we have now also included in the revised manuscript (**Supplementary Table 2**).

Substrate (200 nM)	MmATD	k_{obs} (min ⁻¹)*	$k_{obs}/[\text{Enzyme}]$ (min ⁻¹ nM ⁻¹)
L-Tyr-tRNA^{Tyr}	5 μ M	No activity	-
D-Tyr-tRNA^{Tyr}	50 nM	0.36 \pm 0.03	0.0072
Gly-tRNA^{Gly}	500 nM	0.55 \pm 0.04	0.0011
L-Ala-tRNA^{Ala}	500 nM	0.15 \pm 0.03	0.0003
L-Thr-tRNA^{Thr}	50 nM	0.19 \pm 0.05	0.0038
L-Ala-tRNA^{Thr}	1 nM	0.27 \pm 0.02	0.27
L-Ala-tRNA^{Cys}	50 nM	0.55 \pm 0.05	0.011

* The deacylation graphs/curves for the calculation of k_{obs} have been taken from Figs. 4 and 5 as well as Supplementary Fig. 6 in the revised manuscript.

4. Figures 4 and 5, the apparent deacylation rate of MmATD is ~500-fold higher for L-Ala-tRNA^{Thr} than for L-Ala-tRNA^{Ala}. Is there any structural model to explain how it recognizes the first few pairs of the tRNA^{Thr}?

Reply:

As is evident from the aforesaid data, the ~500-fold difference in the activity of MmATD on the two substrates, L-Ala-tRNA^{Thr} and L-Ala-tRNA^{Ala}, is attributed to the differences in the two tRNAs, viz., tRNA^{Ala} and tRNA^{Thr}. Hence, the focus of our future work will be to identify the elements of the tRNA that are involved in this discrimination and the underlying mechanism through biochemical, biophysical and structural investigations, including the determination of ligand-bound crystal structures. This will allow us to propose a structural/mechanistic model for ATD's substrate specificity.

5. Has it been tested to see if L-Ala-tRNA^{Thr} is a substrate for DTD?

Reply:

We are grateful to the reviewer for bringing up a relevant aspect of DTD with regard to its activity on L-Ala-tRNA^{Thr}. We have now tested DTD's activity on L-Ala-tRNA^{Thr} and we show that the enzyme does not act on the substrate even at 100 nM concentration (**Fig. 2**) compared to ATD's activity on the same substrate at 1 nM concentration (**Fig. 1a**). The data have now been included in the revised manuscript (**Supplementary Fig. 7**).

Fig. 2 Deacylation of L-Ala-tRNA^{Thr}(G4•U69) by MmDTD.

DTD's inactivity on L-Ala-tRNA^{Thr} is expected, since L-chiral rejection is the only design principle through which it achieves its remarkable enantioselectivity (**Ahmad et al., eLife, 2013; Routh et al., PLoS Biol, 2016**). It employs the parallel orientation of carbonyl oxygens of Gly-cisPro motif which protrude into the active site to sterically exclude even the smallest amino acid with L-chirality, i.e. L-alanine. Thus, the chiral proofreading enzyme

acts on D-Tyr-tRNA^{Tyr} and Gly-tRNA^{Ala} at picomolar concentrations (**Ahmad et al., eLife, 2013; Pawar et al., eLife , 2017**), but fails to act on L-Tyr-tRNA^{Tyr} and L-Ala-tRNA^{Ala} even at high-nanomolar to micromolar concentrations (**Ahmad et al., eLife, 2013; Routh et al., PLoS Biol, 2016**). Interestingly, such an exclusionary mechanism to attain substrate chiral specificity by DTD does not involve any tRNA element as the active site pocket discriminates between the substrates exclusively on the basis of chirality of the aminoacyl moiety. Taking all the above observations into consideration, it was highly unlikely for DTD to act on L-Ala-tRNA^{Thr}. Nevertheless, it was important to test DTD's activity on L-Ala-tRNA^{Thr} to rule out any role of the enzyme in editing this non-cognate species.

6. *Line 230 and 640, it's too strong a statement to claim that EF-Tu does not protect L-Ala-tRNA^{Thr} from ATD. In fact, Figures 6A and 6B shows that at 5 nM ATD, EF-Tu shows some protective effect. The authors need to revise the statement and related discussion.*

Reply:

As correctly pointed out by the reviewer, we have now revised the statement and related discussion in question, thereby removing the discrepancy between the data and their inference.

Reviewer 3

No comments from the reviewer